# Rapid and Nondestructive Classification of Cantonese Sausage Degree Using Hyperspectral Images

**Qi Wang [1] and Yong He [1,2,3,\*]**

[1] College of Biosystems Engineering and Food Science, Zhejiang University, Hangzhou 310058, China; 3120100288@zju.edu.cn

[2] State Key Laboratory of Modern Optical Instrumentation, Zhejiang University, Hangzhou 310058, China

[3] Key Laboratory of Spectroscopy Sensing, Ministry of Agriculture and Rural Affairs, Hangzhou 310058, China

\* Correspondence: yhe@zju.edu.cn; Tel.: +86-571-8898-2143



**Featured Application: The work used hyperspectral image technology to model the quality grade of Cantonese sausage, which can be used for non-destructive detection in sausage processing and production.**

**Abstract:** Hyperspectral images are widely used in the food industry as a fast and non-destructive analytical technique. Cantonese sausage has a long history and is a very old food production and meat preservation technology. According to the physical and chemical characteristics of the sausage, the Chinese business industry standard SB/t10003-92 divides the sausage into three grades, which are called excellent grade, level 1, and level 2. In this paper, k-means is adopted first to separate two parts of the meat adaptively to improve the discriminant rate. The hyperspectral information of the near-infrared band is extracted by successive projections algorithm (SPA). The multiple linear regression (MLR) and partial least squares regression (PLSR) algorithms are used to classify the sausage grade. The experimental results show that the lean meat and fat of the sausage have different characteristics in the near-infrared band, and the modeling results have higher accuracy and anti-interference after separating lean meat and fat meat. The best model of sausage classification is using SPA-MLR method to model the fat region of Cantonese sausage; the prediction accuracy of which is 100%. It was found that the modeling results of fat were better than lean meat in both PLSR and SPA-MLR, which indicated that there were obvious differences in fat composition among different grades of sausage, and the fat of sausage was more suitable for classification.

**Keywords:** Cantonese sausage; near infrared spectroscopy; k-means; PLSR; SPA

## 1. Introduction

Chinese sausage is a kind of meat product with Chinese characteristics which refer to the meat as raw material, which is broken into small pieces that, together with the excipients, is poured into the animal casings for fermentation and converted to dry meat products. Sausage is one of the largest varieties of Chinese meat products, and the main origins of sausage include China's Guangdong, Zhejiang, Shandong, and other places, of which Cantonese sausage is one of the most famous. With the exception of slightly different materials, the production method of all varities of sausage is roughly the same. Therefore, the non-destructive detection of sausages is of great significance to the quality identification and the protection of trademark rights.

The Cantonese sausage grade identification method requires the detection of some of the major components in the sausage, including protein, oil, acid value, nitrite, POV(peroxide value), and so on.

Every measurement involves a number of chemical reagents, and the measurement cycle is long. Traditionally, some measurements use the national standard method to measure the physical and chemical properties of sausage. In the study of the effects of Flavourzyme on proteolysis, antioxidant capacity, and sensory attributes of Chinese sausage, ISO-2917 method was used to determine the moisture content, water activity, pH of the sausage, and the protein [1,2]. Both methods need to dissolve the sausage, and also involve complex pretreatment. Acid and peroxide value can be regarded as detection indices of Cantonese sausage which is stored with changes to the physical and chemical characteristics, such as baking, traditional sun, and so on [3]. The acid value is measured by crushing the sausage and dissolving it with petroleum ether first and then titrating free fatty acid content by potassium hydroxide. Pretreatment is complex and time-consuming, and extensive use of chemical reagents leads to environmental pollution. At the same time, these methods cannot achieve rapid, accurate and comprehensive quality identification of sausage.

Gas chromatography–mass spectrometry (GC-MS) is a commonly used method for the determination of components in food flavor substances [4,5]. It is suitable for the analysis of complex environments containing many organisms. For meat products, GC-MS can effectively detect organic compounds, such as aldehydes, esters, and so on [6,7]. And the solid phase micro extraction process (SPME)-GC-MS method is usually used to explore volatile components in naturally dried sausage [8]. First, the volatile ingredients were extracted with SPME and then identified with GC-MS. The results showed that the main volatile substances in sausage were hexanal, ethyl, 2-pentanone, D-limonene, and hexanoic acid ethyl ester. The formation of these volatile substances may be due to of fat oxidation, alcohol, microbial activity, spices, and the interaction among them. GC-MS has a high degree of precision, but the detection steps are complex and costly, and it is difficult to extract samples for unevenly distributed mixtures and is, therefore, not suitable for practical production.

Traditional chemical detection methods are complex and time-consuming, and extensive use of chemical reagents leads to environmental pollution. At the same time, these methods cannot be fast and accurate for sausage quality identification. Near-infrared spectroscopy is the most commonly used nondestructive testing technique in food detection and classification because it does not require direct contact with the measured object [9–12]. It can accurately determine the composition and content of fat, protein, and water in meat [13–15]. Gaitán-Jurado used near infrared spectroscopy (NIR) to model fat, moisture, and protein in smoked meat sausage which is pretreated by slicing and crushing [16]. The correlation coefficients of fat, moisture, and protein in sausage processed by slicing were 0.98, 0.93, 0.97. Liao Yi-Tao used visible/near-infrared technology to detect fresh pork by constructing the detection system, and the establishment of the partial least squares regression model predicted the correlation coefficient of intramuscular fat, protein, and water content $\geq$0.81 [17]. In addition to chemical composition, sensory evaluation is also an important indicator of meat quality, including color, texture, flavor, and juiciness, etc. Due to the relationship between these indicators and the physical and chemical properties, sensory evaluation of meat products can be detected by near-infrared spectroscopy. Prieto studied the pH, color attributes, and hydraulic system of adult beef and juvenile beef by near-infrared spectroscopy (1100–2500 nm) [18]. It was found that the correlation coefficients of L* and b* in color characteristics were 0.869 and 0.901, respectively. Ellekjaer using NIR (near infrared spectroscopy) assessed the sensory quality of meat sausages. The NIR spectra described the main color and texture variations among sausages with a relative prediction of 0.87 to 0.93 for color attributes and 0.83 to 0.91 for juiciness and greasiness [19]. Hyperspectral imaging spectroscopy can also be used to monitor changes in physicochemical, microbiological, and sensory attributes of the packaged dry-cured sausages, and the developed technique can be implemented in quality inspection of food products without additional laborious chemical analysis and sensory evaluation [20].

All of the above research methods average hyperspectral data of each sample, which represents the spectral information of one sample, and ignores the component differences in different parts of meat, so it does not make full use of hyperspectral image information. The purpose of this paper is to (1) identify the grade of Cantonese sausage by hyperspectral image information, (2) find the effective

wavelengths of the sausage samples in NIR bands, (3) compare the discriminant effect of PLSR and SPA-MLR, and (4) analysis the correlation between sausage grade with lean meat or fat meat.

## 2. Materials and Methods

### 2.1. Sausage Sample

In this experiment, a total of eight batches of Cantonese sausage were purchased, of which twenty-two were level 2 sausages, twenty-nine were level 1 sausages, twenty-nine were excellent grade sausages, from "XiShangXi" and "HuangShangHuang" which are two well-known Cantonese sausage brands. Each sausage was about 17 cm long and 1.5 cm in diameter. These sausages were stored in a refrigerator at minus 30 °C. Before the experiment, sausage samples were taken from the refrigerator and put on fume hood for half an hour to reach laboratory temperature (20 ± 1 °C). Different grades of sausages were organised in batches to prevent surface grease contact with each other. This operation can also evaporate water vapor from the surface.

### 2.2. Hyperspectral Imaging System

The hyperspectral images of the samples were obtained by the reflectance mode of a laboratory-based pushbroom hyperspectral imaging system which records the hyperspectral information of a two-dimensional matrix. The hyperspectral imaging system consists of a high-performance CCD (charge coupled device camera) camera, a mobile platform used for samples shifting, an imaging spectrograph (ImSpectorV10, Spectral Imaging LTD., Oulu, Finland) and a computer supported with Spectral-Cube data acquisition software (Spectral Imaging Ltd., Oulu, Finland,2007) which controls the motor speed, exposure time, binning mode, wavelength range and image acquisition. An illumination unit containing two 150 W quartz tungsten halogen lamps (Oriel Instruments, California, USA) was fixed above the mobile platform at 300 mm ground height. The camera spectral range was from 874 nm to 1734 nm divided into 256 bands. The camera had 672 × 256 (spatial × spectral) pixels with a spectral resolution of 2.8 nm. UnscramblerX 10.1 (CAMO Process as, Oslo, Norway, 2004) software, ENVI 5.2 (Research System Inc., Boulder, CO, USA, 2014) and MATLAB R2016b (The MathWorks Inc., Natick, MA, USA, 2016) were used to analyze the spectral data in the study. Figure 1 shows the process of hyperspectral image acquisition in our experiment.

### 2.3. Hyperspectral Image Acquisition

During the data collection phase, the sausage samples were taken out from the fume hood and immediately put into the hyperspectral imaging system for image acquisition. Each of the three samples was placed on a mobile platform as a group, and the distance from the samples to the lens was about 31.2 cm. The hyperspectral imaging system using 3 ms exposure time, which records a whole line of an image rather than a single pixel at every turn. The velocity of the mobile platform was 2.6 cm/s, and the dimension of the hyperspectral image is (x, y, λ), where (x, y) is the spatial size of the hyperspectral image. In this experiment, there were 672 pixels in x-direction, and the number of pixels in y-direction was due to movement time of the platform, and there was a little difference each time. Each pixel had 256 wavelengths in λ-dimension with 3.37 nm intervals between adjacent bands. To calculate reflectance spectrum, it is necessary to correct the original hyperspectral data by using two reference standards: a white reference image (W) with the spectral reflectance of 99% which was obtained using a Teflon panel, a dark reference image (B) with the spectral reflectance of 0% which was acquired by turning off the light source and covering the lens. For the noise generated in the blackboard correction, the software denoising algorithm was used first to get the spectrum to 0% reflectivity. Finally, the calibrated image (I) can be deduced from the original hyperspectral image ($I_0$) using the following equation:

$$I = \frac{I_0 - B}{W - B} \tag{1}$$

The calibrated image was the basis of the subsequent image analysis, and then effective wavelengths could be extracted and analyzed.

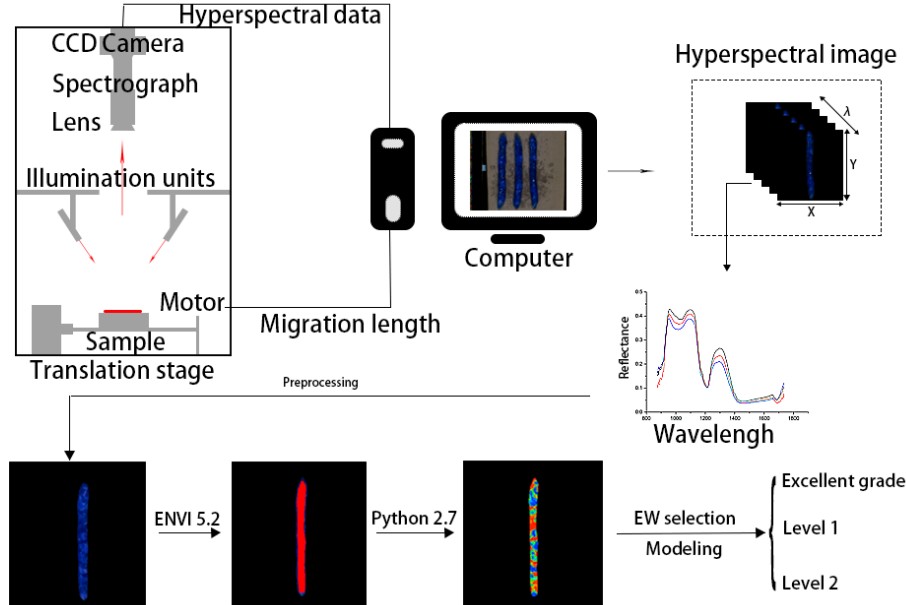

**Figure 1.** The flow chart of hyperspectral detection and image processing; The CCD camera was located at the top of the translation stage, through which the sausage hyperspectral image was acquired and transmitted to the computer. Then ENVI was used to visualize hyperspectral images and extract ROI (region of interesting). Finally, Python 2.7 was used to select effective wavelengths (EW) and model them.

## 2.4. Image Preprocessing

Before spectral analyzing, the region of interest (ROI) needs to be manually selected from the background first. The smooth outer skin of the sausage can cause several exposure points, thus, removing these points is necessary for improving the modeling accuracy. Then, the same routine was repeated for hyperspectral images of all samples. The size of each samples' ROI was about 4000 pixels. All 80 ROIs' image were obtained from 80 samples, in which 54 samples were selected as training sets randomly, and the remaining 26 samples were treated as the prediction set.

### 2.4.1. Image Repartition

The k-means algorithm belongs to cluster analysis, which is used to divide a set of objects into several groups according to the similarity of each other, where the similar objects constitute a group. The k-means algorithm has proved useful in clustering and segmentation of hyperspectral images [21–23].

It was obvious from the hyperspectral image that the sausage is mainly composed of two parts, lean meat and fat, and there were significant differences between the two parts due to chemical composition reflected in different hyperspectral images. So, if all the pixels of each ROI were averaged to generate one mean value in each band, a significant quantity of potential information would be lost.

By using the k-means adaptive segmentation algorithm, all the pixels in each ROI were divided into two categories automatically, which can reduce the deviation from the judgment of human eyes, save time for manual segmentation, and allow for mining of more high-dimensional image information. As lean meat is always the main component of the sausage, the class with larger pixel number belongs to lean meat, and the other is fat. Through the k-means clustering algorithm, the categories of each pixel in the ROIs and the clustering centers of two classes were obtained. As the sausage samples of

lean meat and fat often mixed together, the high spectral values of each pixel point represent the result of lean meat and fat's mix in different proportions. Therefore, the membership degree is introduced to evaluate the proportion of lean meat and fat to each pixel. Referencing the gravitational model, we call the similarity of samples compared to lean meat and fat in each pixel $R_L$ and $R_F$, and the definition is:

$$R_L{}^2 = \frac{1}{\left(H_{pix} - H_{L-mean}\right)^2} \tag{2}$$

$$R_F{}^2 = \frac{1}{\left(H_{pix} - H_{F-mean}\right)^2} \tag{3}$$

where $H_{pix}$ is the vector of spectral values in a pixel, $H_L$ is the average spectral values' vector of lean meat class, and $H_{F-mean}$ is the average spectral values' vector of fat class.

Through the above formula, we can deduce the membership degree of lean meat and fat of each pixel in ROIs as follows:

$$M_L = \frac{R_L{}^2}{R_L{}^2 + R_F{}^2} \tag{4}$$

$$M_F = \frac{R_F{}^2}{R_L{}^2 + R_F{}^2} \tag{5}$$

where $M_L$ is the membership degree of lean meat in a pixel, and $M_F$ is the membership degree of fat in a pixel. It is obvious that the sum of $M_L$ and $M_F$ of the same pixel is one.

For each sample, the hyperspectral information corresponding to the lean meat ($I_L$) is obtained according to the weight average of the lean meat membership degree, and so is the fat ($I_F$). Figure 2 shows the segmentation effect of some samples.

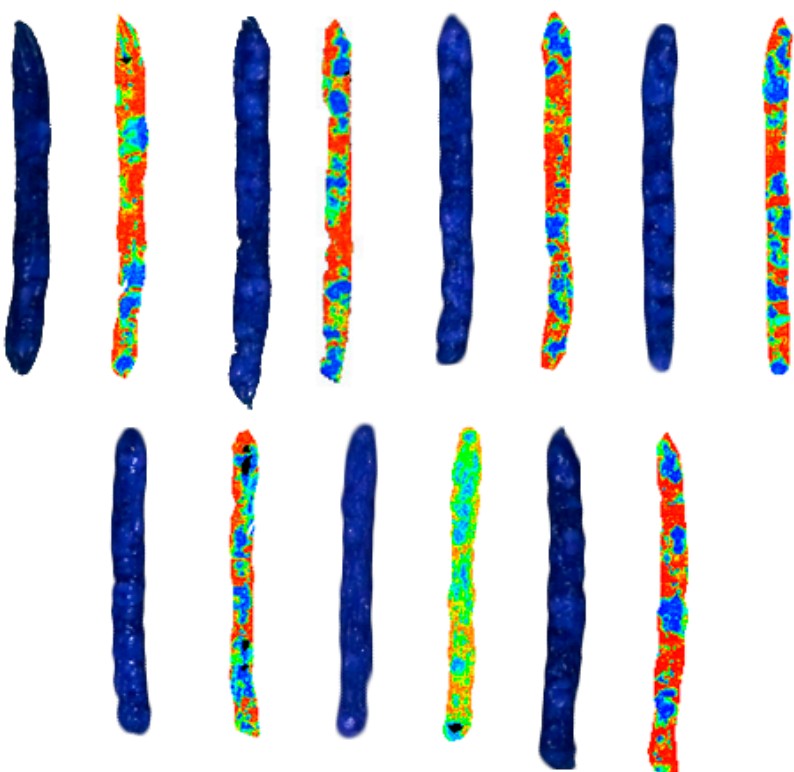

**Figure 2.** Seven groups of near infrared spectroscopy (NIR) hyperspectral images of sausage samples and the corresponding pseudo color images. The deeper the red, the higher the degree of lean meat membership.

### 2.4.2. PLSR Modeling Method

Partial least squares regression (PLSR) is a multivariate data analysis method based on principal component analysis and principal component regression. The salient feature of partial PLSR is the effect of the overall effect of the sample on the prediction, and the effect of the combined effect of the factors on the predicted value. On the basis of the image, the effective hyperspectral information of the sausage samples was obtained by PLSR, and the correlation between the hyperspectral data and the sausage grade was established. The evaluation index of the model are the root mean square error (RMSE), coefficient of determination($R_2$), and the grade judgment. The smaller the root error and the larger the $R_2$, the better the prediction effect of the model.

The following standards can be regarded as the predictive performance of three regression models: root mean square error of training set (RMSEC) and prediction set (RMSEP), coefficient of determination in training set ($R_c^2$) and prediction set ($R_p^2$), discrimination rate of training ($D_c$) and prediction ($D_P$). The discriminant rate is obtained from the regression model which predicts the ratio of the number of correct classifications in the number of total predicted set, and the category in (1, 2, 3) which is closest is taken as the predicted value of the sample which represent excellent grade, level 1, and level 2 respectively.

### 2.4.3. SPA-MLR Modeling Method

The successive projections algorithm (SPA) can find the set of variables containing the minimum redundant information from the spectral information fully [24,25] so that the collinearity between the variables can be minimized and greatly reduce the number of variables used in modeling to improve the speed and efficiency of modeling.

The hyperspectral value of the whole sausage was obtained by the K-means algorithm, and then for $I_L$ and $I_F$, the effective wavelengths were respectively extracted from 256 NIR bands using SPA. Finally, multiple linear regression (MLR) was used to model the correlation between the reflectance spectra and sausage grade by only referencing effective wavelengths as the feature of each sample.

## 3. Results

### 3.1. Spectral Characteristics of Sausage

Figure 3a contains 80 hyperspectral curves, each corresponding to the average spectrum of a sample. Figure 3b shows the statistical spectrum deduced from the ROI region of all the sausage samples. From the synthesized hyperspectral information, it can be found that the standard deviation (STDEV) in the range of (874, 928 nm) and (1686, 1734 nm) is abnormally large, which means that the signal-to-noise ratio is very low in this region.

The three curves of Figure 3c are the average spectra of three degrees of sausage samples, and the hyperspectral curves of the three classes have the same overall variation trend. In the 800 nm and 1600 nm bands, there are two characteristic peaks, which then fall rapidly and fall to the trough at 1230 nm, and the gradually become stable after the new wave crest at 1300 nm.

Douglas explored the feasibility of using the hyperspectral imaging technique for detecting chemical composition in the pork, and the study indicated that the NIR spectral range had an excellent ability to predict the content of protein, moisture, and fat [26]. Hyperspectral imaging can effectively reflect the protein and oil information of meat, while the oil content can be better predicted than protein by hyperspectral information, and protein and oil are the main components of lean meat and fat meat. By comparing of the three classes of hyperspectral curves it can be found that there is a gradient between the three; the excellent grade sausage spectral reflectivity is highest, the level 2 sausage reflex rate is the smallest, and the level 1 sausage is between the two. Therefore, the increasing sequence (1, 2, 3) was used to label three grades of sausages respectively, which can show the gradient change of characteristics among them. Fifty-four sausages of all eighty samples were used as training sets and the reset as prediction sets.

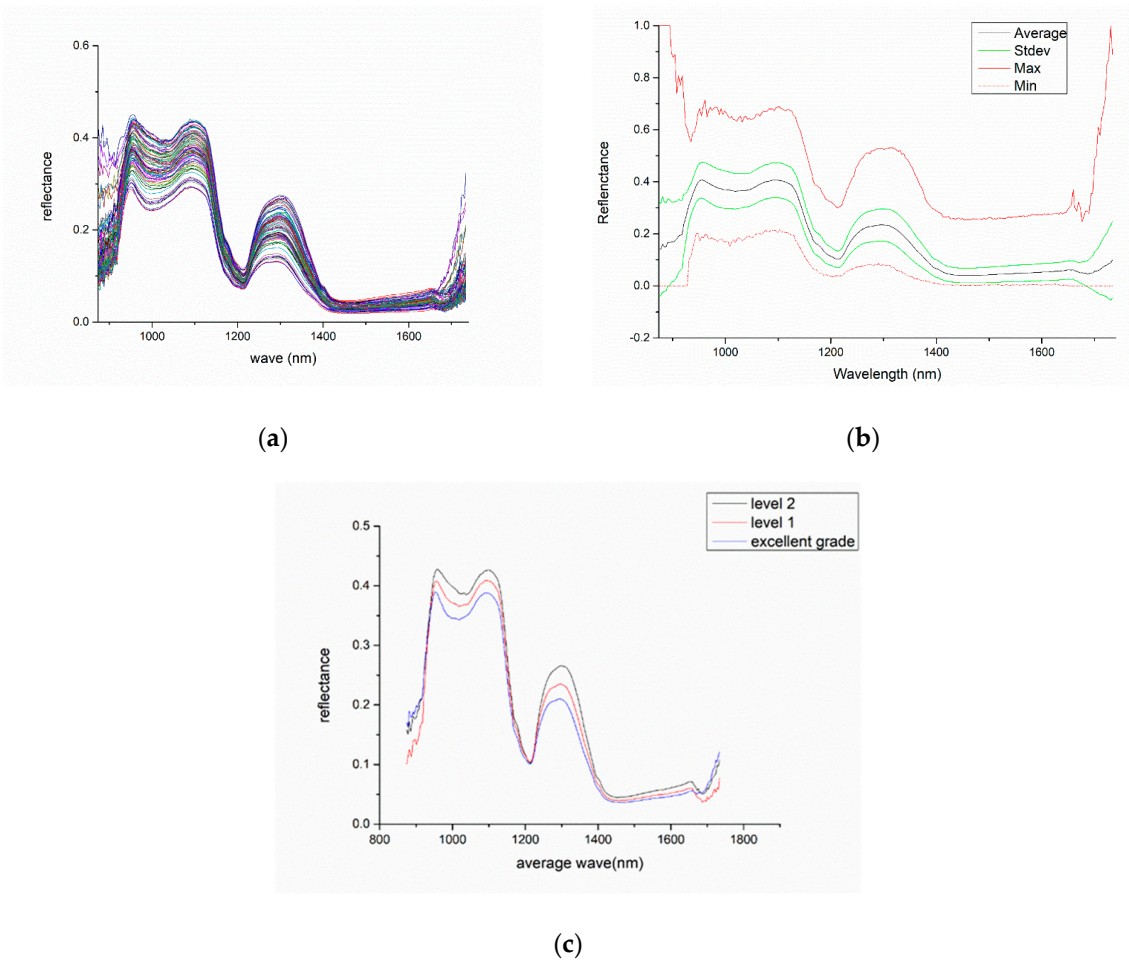

**Figure 3.** (**a**) Average spectrum of region of interest (ROI) for 80 samples in NIR bands (874–1734 nm); (**b**) NIR reflectance spectra of all 80 sausage samples, the red line and yellow line are the maximum and minimum reflectivity of all samples corresponding to each wavelength, and the green line represents the standard deviation of the corresponding wavelength, which can reflect the effective information contained in each wavelength; (**c**) Average hyperspectral reflectance of sausage samples in three degrees.

### 3.2. Prediction of Sausage Degree using All Wavelengths

#### 3.2.1. Noise Band Removal

The STDEV in the middle part of the spectrum (928–1686 nm) is below 0.3, which indicates that the reflection of these bands changes slightly with individual differences and can reflect the chemical composition of the sausages more stably. As can be seen from Figure 3c, there are significant differences in the reflectance among the three degrees at four peaks (955, 1600, 1300, 1670 nm). Finally, 224 bands from 928 to 1686 nm were selected as characteristics wavelengths for full-band modeling.

#### 3.2.2. Modeling Results of PLSR

Three models were established by the PLSR method, which corresponded to non-segmented, lean, and fat meat samples, respectively. For the sample without repartition, the first four loading weights cover the over 90% of information of spectral variances, so the top four loading weights are used. It can be seen in Figure 4. The same treatment is made for the modeling of lean meat and fat meat. Fifty-four samples were used for modeling and 26 samples for prediction. The results were shown in the following Table 1.

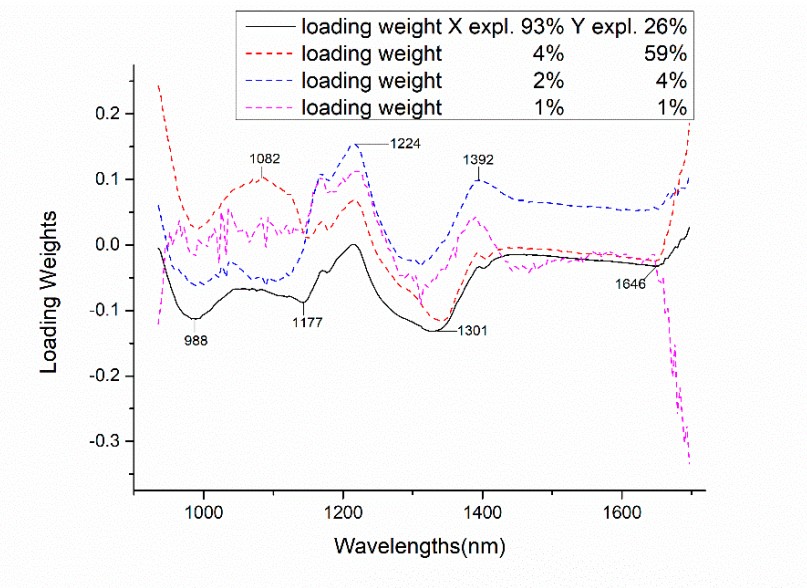

**Figure 4.** First four loading weights of the best fit partial least squares regression (PLSR) model of no-repartition. The variable X denotes the near-infrared reflectance spectrum; the variable Y denotes the sausage grade.

**Table 1.** Result statistics for Cantonese sausage degree with near infrared spectroscopy (NIR) spectra extracted from samples using the (928–1686 nm) bands from the PLSR models, which include coefficient of determination of the training set ($R_{c2}$), root mean square error of the training set (RMSEC), coefficient of determination of the prediction set ($R_{P2}$), root mean square error of the prediction set (RMSEP), discrimination rate of the training set ($D_c$), discrimination rate of the prediction set ($D_P$).

| ROI | $R_c{}^2$ | RMSEC | $R_P{}^2$ | RMSEP | $D_c$ | $D_P$ |
|---|---|---|---|---|---|---|
| All | 0.9890 | 0.0831 | 0.7775 | 0.1828 | 100% | 100% |
| Lean meat | 0.9881 | 0.0865 | 0.6827 | 0.2209 | 100% | 92.31% |
| Fat meat | 0.9875 | 0.0885 | 0.8991 | 0.1996 | 100% | 100% |

From the data above, we can see that the $R_c{}^2$ of the training set are almost the same, but the $R_P{}^2$ on prediction sets show a gradient change. The modeling results of no repartition sets are centered, and the $R_P{}^2$ of fat meat part is 0.8991, compared with the 0.6827 of lean meat part. This indicates that there is a high correlation between the fat fraction and the sausage grade in the sausage samples, and a large difference exists in the chemical composition of sausages' fat fraction between different grades, while the correlation of the lean meat section is weak.

### 3.3. SPA-MLR

#### 3.3.1. Effective Wavelengths Selection

Due to the high correlation between adjacent bands of hyperspectral data, there is siginficant information redundancy in full-band modeling. Therefore, modeling using effective wavelengths is more effective than full band. In this study, SPA is used to extract the characteristic band, and then the characteristic band is modeled by MLR. The Savitzky–Golay (S-G) filter is a commonly used low-pass signal within the time domain filter. It is suitable for smoothing continuous signals and has the characteristics of fast calculation and strong operability [27]. Thus, the hyperspectral data preprocessed by S-G filter can effectively utilize the information in adjacent spectra with less noise. The parameter polynomial order of the S-G filter is two, the frame length is 15, and the preprocessed characteristic spectrum of 242-dimensional is obtained. The head and tail part of the spectrum is

truncated in preprocessing, but considering that the head and tail contain significant noise, truncation of these two parts does not have much effect on the modeling accuracy.

Ten effective wavelengths were extracted from the hyperspectral data of 80 samples without repartition by SPA, which contained (897.87, 907.93, 921.34, 1008.58, 1095.92, 1213.65, 1341.69, 1598.43, 1683.07, 1710.18 nm). After the separation of lean meat and fat meat by k-means, the effective wavelengths were extracted by SPA respectively. The effective wavelengths of lean meat contained (907.93, 911.28, 921.34, 934.76, 968.30, 1022.01, 1116.09, 1156.44, 1338.32, 1422.67, 1588.28, 1710.18 nm), and the other contained (897.87, 907.93, 917.99, 951.53, 1082.47,.1163.9, 1304.6, 1453.06, 1588.28 nm). It was found that there were common selected bands in two groups, such as 907.93 nm, 1588.28 nm. The reason is that lean meat and fat meat are mainly composed of C, H, O elements, and the C=O and C-H formed by them are abundant in protein and fat. However, all the selected wavelengths of the two parts had a large difference which reflected the difference between the lean meat and fat meat.

### 3.3.2. Regression Analysis

For the three sets of selected wavelengths, MLR was used to carry out regression modeling. Fifty-four sausages samples were used for modeling and 26 samples for prediction. The modeling results are shown in Table 2.

**Table 2.** Multiple linear regression (MLR) modeling result statistics for Cantonese sausage degree with NIR spectra using selected wavelengths extracted by successive projections algorithm (SPA).

| ROI | $R_c{}^2$ | RMSEC | $R_P{}^2$ | RMSEP | $D_c$ | $D_P$ |
|---|---|---|---|---|---|---|
| All | 0.8656 | 0.1635 | 0.8387 | 0.2341 | 100% | 96.15% |
| Lean meat | 0.9371 | 0.1635 | 0.9153 | 0.2422 | 100% | 100% |
| Fat meat | 0.9320 | 0.1686 | 0.8955 | 0.2619 | 98.15% | 96.15% |

As can be seen from the table, both the lean meat and fat meat sets have a high $R_c{}^2$ by using SPA modeling, while the $R_c{}^2$ of the no-repartition sets is low. The same feature is shown in the prediction sets. After the repartition of sausage samples, the use of SPA for the selection of the effective band can use a small amount of band to extract a larger amount of information. It indicated that the distribution of effective information in the spectrum is more concentrated, and also proved the effectiveness of repartition. All show a high accuracy.

From the discrimination rate results, lean meat set and no-repartition sets both have one misjudgment sample, while the $D_P$ of fat meat set is 100%.

## 4. Discussion

The study illustrates the effectiveness of rapid, nondestructive detection of sausage grades using the NIR bands at (874–1734 nm). Intact and sliced sausages were used to explore the feasibility of using hyperspectral imaging to grade Cantonese sausages in some studies [28,29]. It was found that intact sausages had better classification effect than sliced sausages. In fact, there is a layer of intestinal skin outside intact sausages, which will interfere with the acquisition of spectrum, and sliced sausages can avoid this problem, so the result of sliced sausages should be better. However, the experimental results were different. The explanation is that the ROI of sliced sausages samples is small and variance of the proportion of lean and fat meat is large which have a great influence on the average spectrum. On the contrary, intact sausages are less affected, so the robustness of the model is better than that of sliced sausages. In this study, the k-means algorithm was used to segment lean meat and fat meat regions before modeling, which avoided the influence of the proportion of lean and fat meat and improved the robustness of the model. Preliminary separation of fat meat and lean meat was carried out by k-means, and then PLSR and SPA-MLR modeling of two separated sets and no-repartition sets were performed, respectively. Among them, the prediction of fat meat under SPA-MLR has the highest coefficient of decision ($R_P{}^2 = 0.9153$). At the same time, all the calibration and prediction of the no-repartition set

and fat meat set under PLSR and the fat meat set under SPA-MLR reached 100%. From the results of the lean meat and fat meat sets modeled by PLSR, it was found that the fat meat in sausage reflected the quality grade of the sausage better than that of lean meat. According to the effective wavelengths extracted by SPA and load weightings of PLSR, we can see that the main bands of effective information are distributed around 950 nm, 1150 nm, 1300 nm, and that there are significant differences in the reflectance spectra of different grades of sausages. This study had proved that nondestructive detection using hyperspectral information can replace traditional chemical detection methods and achieve rapid classification of sausage grades. At the same time, the robustness of the model can be further improved by separating lean meat and fat.

**Author Contributions:** Conceptualization, Q.W. and Y.H.; methodology, Y.H.; software, Q.W.; validation, Q.W. and Y.H.; formal analysis, Y.H.; investigation, Q.W.; resources, Y.H.; data curation, Q.W.; writing—original draft preparation, Q.W.; writing—review and editing, Y.H.; visualization, Q.W.; supervision, Y.H.; project administration, Y.H.; funding acquisition, Y.H.

**Funding:** This research was funded by National key R&D program of China 2018YFD0101002.

**Conflicts of Interest:** The authors declare no conflict of interest.

## Abbreviations

The following abbreviations are used in this manuscript:

| | |
|---|---|
| SPA | successive projections algorithm |
| PLSR | partial least squares regression |
| NIR | near infrared spectroscopy |
| MLR | multiple linear regression |
| ROI | region of interest |
| RMSE | root mean square error |
| STDEV | standard deviation |
| SPME | solid phase micro extraction process |
| GC-MS | gas chromatography-mass spectrometry |

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
