# Peer review of "Rapid and Nondestructive Classification of Cantonese Sausage Degree Using Hyperspectral Images"

_applsci, doi:10.3390/app9050822_

Round 1
Reviewer 1 Report
Dear Authors, Your research is of high practical relevance with industrial interest making the project very realistic. Introduction part is well written; the ending part gives a clear overview of what will be focused on, with clarified goals. Line 126-131: You write about reference panels and measurements. There it is described that your white reference is 99% reflective, which is very high. This is an averaged value but changing for each wavelength. Did you get a correction curve with your Teflon? If you have this you can enhance the spectral quality and accuracy because each wavelength will be considered and can be corrected, not only one average for every wavelength. Do you have any information on this? Another issue is that black reference was made by closing the optical input, which helps find internal noise. If it is noisy, in most cases it is, removing has to be done. Did you do or control this? Line 131: Instead of (I0) correct for (I0). Line 232: You missed the initial letter. Figures 3 and 4 should be put together because they are taking too much place and do not give proportionally more information. I miss some basic information in your statistics, for instance. Number of samples (n), which are typically given in statistical results´ table such as Table 1. and Table 2. Generally it is a nice work for industrial applications this is a good collection of proven methods; it has more an application note character than a research paper. Your statistics should be more precise and detailed.Author Response
Thank you for your detailed and patient advice, which is very meaningful for my research. I summarized your comments in word file and made corresponding revisions in the original paper.

Reviewer 2 Report
Manuscript ID: Applsci-447617
Title: Rapid and nondestructive Classification of Cantonese Sausage Degree using Hyperspectral Images.
The purpose of the revised manuscript is to solve an interesting problem in the Cantonese sausage market: the rapid and non-destructive discrimination of the different qualities of sausages. It pretends to be a complete and useful study. However, there are some important concerns that require a careful and thorough review before its hypothetical publication.
Introduction
In general, the Introduction section is clear and precise. It stablishes the importance of the product and the existence of different grades. In addition, it describes hyperspectral imaging and resumes the use of this tool in sausage. However, the introduction section forgets the three most similar papers that can be easily found in a quick bibliographic search ... These studies are:
- Hyperspectral imaging for rapid evaluation and visualization of quality deterioration index of vacuum packaged dry-cured sausages. Siripatrawan, Ubonrat, SENSORS AND ACTUATORS B-CHEMICAL. Vol: 254 Pag: 1025-1032. JAN 2018.
- Grading of Chinese Cantonese Sausage Using Hyperspectral Imaging Combined with Chemometric Methods. Gong, Aiping; Zhu, Susu; He, Yong; et ál.. SENSORS. Vol: 17, N: 8, Paper Number: 1706. AUG 2017
- Study on the Quality Classification of Sausage with Hyperspectral Infrared Band. Gong Ai-ping; Wang Qi; Shao Yong-ni. SPECTROSCOPY AND SPECTRAL ANALYSIS. Vol.: 37 N: 8 Pag: 2556-2559, Paper Number: 1000-0593(2017)37:8<2556:LYGGPJ>2.0.TX;2-A. AUG 2017.
Given that the authors of this manuscript are co-authors of two of the previous references, I do not understand why they did not include them as a bibliography and commented on them in the introduction section. I can only think that they wanted to hide them so as not to reveal the similarity with the current manuscript. They should include these references and highlight the differences between these documents and the current one.
However, even taking into account the presence of these works, I think that the current manuscript can be interesting for publication in Applied Sciences after major revision.
Lines 88-89: Mainly the aims 1 and 2 are really similar to those addressed in the previous studies. Please, clarify why the current aims are different than the older.
Material and Methods
Lines 94-95: Do you think that only 80 samples are enough for achieving your aims? Justify why did you use such a small number and why this number is enough for achieve your aims.
Lines 139-140: Did you take into account the different grades of sausage to select the training and prediction set? Or the selection was completely random?
Results and discussion
Please, carefully check the grammar and style of the entire manuscript, but pay special attention to this section. There are several grammatical errors, most of them confusing the order of the words.
Line 201: STDEV, If this is an abbreviation for standard deviation I did not know it... It is better to add the complete term and add the abbreviation to the abbreviation list at the end of the manuscript.
Caption of Figure 3 (b): Principal statistical descriptors of NIR reflectance spectra of...
Line 205: What do you mean with synthetic? In my opinion that is a confusing term. Please replace it.
Lines 206-208: I think that in this sentence there are several grammatical errors and moreover errors in the band numbers. Check it please.
Lines 215-217, 225-228, 232, 240: I saw some grammatical errors in these sentences. Please reread the entire manuscript.
Line 227: The peaks described in brackets do not correspond with those highlighted in Figure 5. Please, check it.
Figure 5: What means X and Y? I do not find that in the manuscript.
Figure 5: The variability explained by Y adds 90%. Should not it add up to 100%?
Lines 243-244: Chemical composition or amount of fat in the sausages? It could be logical a relationship between the amount of fat and the grade of the sausage...
Line 266: I think is not correct to say ingredients in that context... If the same peaks are important for both matrixes, it only wants to say that the bonds that vibrate at those wavelengths are present in both matrixes. It could be interesting to search in bibliography what overtones or combination bands are present at those bands and what compound or compounds could be the responsible.
Line 267: Chemical composition is a term much better than ingredient in that context!
References
Some journal names are abbreviated and some are not.
Author Response
Thank you for your detailed and patient advice, which is very meaningful for my research. I summarized your comments in word file and made corresponding revisions in the original paper.

Round 2
Reviewer 1 Report
Dear Authors,
Thanks for your corrections. I saw you made many changes. But there are still some typos.
For instance, please check:
Line 312: You wrote "We can found...." it is not correct, please check all carefully
Line 333: "in gong´s study".... the same problem again
Please ask a proof reader for checking
Reviewer 2 Report
All the proposed corrections have been taken into account. Therefore, I suggest accepting the work in its current version.